# Properties of Nanohybrid Dental Composites—A Comparative In Vitro Study

**DOI:** 10.3390/biomedicines12010243

**Published:** 2024-01-22

**Authors:** Mihaela Păstrav, Ovidiu Păstrav, Andrea Maria Chisnoiu, Radu Marcel Chisnoiu, Stanca Cuc, Ioan Petean, Codruța Saroși, Dana Feștilă

**Affiliations:** 1Department of Orthodontics, “Iuliu Hațieganu” University of Medicine and Pharmacy, 400349 Cluj-Napoca, Romania; mihaela.pastrav@umfcluj.ro (M.P.); dana.festila@umfcluj.ro (D.F.); 2Department of Odontology, Endodontics and Oral Pathology, “Iuliu Hațieganu” University of Medicine and Pharmacy, 400349 Cluj-Napoca, Romania; ovidiu.pastrav@umfcluj.ro; 3Department of Prosthodontics, “Iuliu Hațieganu” University of Medicine and Pharmacy, 400349 Cluj-Napoca, Romania; 4Raluca Ripan Institute for Research in Chemistry, Babeș-Bolyai University, 400294 Cluj-Napoca, Romania; stanca.cuc@ubbcluj.ro (S.C.); codruta.sarosi@ubbcluj.ro (C.S.); 5Faculty of Chemistry and Chemical Engineering, Babeș-Bolyai University, 400028 Cluj-Napoca, Romania; ioan.petean@ubbcluj.ro

**Keywords:** nanohybrid composites, filler, color stability, roughness, dentistry, restorative dentistry

## Abstract

(1) Background: the current study investigated three nanohybrid composites: two commercial products ClearfilMajesty^TM^ (CM) and Harmonize^TM^ (HU), compared with an experimental product PS2. (2) Methods: Two sample types were molded using Teflon dies. The first sample type was represented by standard discs (20 mm diameter and 2 mm thickness) (*n* = 60, 20/each material), used for surface conditioning investigation, specifically roughness monitoring and color stability analysis using AFM and the CIELab test, respectively. The second sample type was a standard cylindrical specimen (4 mm diameter and 6 mm height) for compression testing (*n* = 60, 20/each material). After complete polymerization, the samples were ground with sandpaper and further polished. The filler size and distribution in the polymer matrix were investigated with SEM. Data were statistically analyzed using the Anova Test followed by Tukey’s post hoc test on the Origin Lab 2019 software produced by OriginLab Corporation, Northampton, MA, USA. (3) Results: A mono-disperse system was identified in HU samples, while CM and PS2 revealed both nano- and microfiller particles. The samples’ observation after immersion in coffee and tea indicated that a lower roughness combined with optimal filler lamination within the polymer matrix assured the best color preservation. The compression strength was lower for the HU sample, while higher values were obtained for the complex filler systems within CM and PS2. (4) Conclusions: the behavior of the investigated nanohybrid composites strongly depends on the microstructural features.

## 1. Introduction

Composite materials are versatile in dentistry applications due to the combined properties exerted by the synergistic action of the polymeric base embedding filler particles [1,2]. The polymeric matrix type plays an important role in the filler embedding, assuring an optimal wetting of the particles and generating a proper lamination of these structures, and facilitates their further modeling into the desired shape prior to the photo-polymerization process. Thus, resins are widely used in dentistry because of their high binding ability and good mechanical properties [3,4]. The addition of filler to the resin matrix generates a composite structure with a certain improvement in mechanical, wear, and optical properties. Thus, resin composites might be designed under desired requirements by properly choosing filler particles (e.g., filler: type, particle shape and size, and its amount) [5,6].

Most of the resins used for dental applications are based on acrylic monomers such as bis-GMA, TEG-DMA, and UDMA [3,7,8,9,10], which are further coupled by silanized agents to form the polymer matrix. Filler particles could be crystalline such as quartz, hydroxyapatite, tri-calcium phosphate, and zirconia [11,12,13] or amorphous such as glass (such as barium oxide or lithium disilicate glasses) [14,15]. Filler particle size is also very important for dental composites whether they are situated in the fine microstructural range of 1–20 µm or are coarse microfillers with particles in the range of 20–100 µm [16]. The newest trends are focused on smaller filler particles situated in the range of 10–99 nm, also called nanoparticles [17]. 

The effect of the filler is the enhancement in the microstructural aspects of the composite that further improves the mechanical properties. Some of the composite resins are designed only with micro- or nanostructured fillers, but the newest trends for dental composites use both micro- and nanofillers, generating complex systems that are defined as nanohybrid structures [18]. The nanohybrid composites are designed for a complex dissipation of the mechanical solicitation within the microstructure to prevent particles delaminating and cracks propagating. This characteristic is sustained by the increased values of the compression and flexural strength of the nanohybrid structures [19,20]. Thus, a uniform filler distribution within the resin base is very important for a homogeneous material with constant mechanical properties. It might be controlled by the proper conditioning of the preparation with respect to the parameters indicated by the manufacturer. On the other hand, their surface conditioning plays an important role in frontal restoration regarding both the surface quality and stain resistance [21]. Literature data show that resin-based composites allow the grading of the color, ensuring an adequate restoration fitted to the tooth shade [18,21]. 

High-resolution imaging is required for the evaluation of micro- and nanostructural characteristics, considering that Scanning Electron Microscopy (SEM) and Atomic Force Microscopy (AFM) represent the most adequate tools for assessment [22]. Micro- and nano-aspects further require a correlation with the mechanical properties. AFM investigations allow proper measurement of the surface roughness, which indicates the quality of surface conditioning [23,24].

The aim of the present study was to investigate three dental composites designed for frontal restorations, a commercial nanohybrid spherical composite, a commercial nanohybrid composite, and an experimental nanohybrid composite, and to reveal their micro- and nano-structural aspects. 

The null hypothesis is that the shape and size of the filler particles do not influence the mechanical properties and surface roughness. 

## 2. Materials and Methods

The current research investigates the conditioning of three dental composites for the frontal restoration of teeth (Table 1): -Clearfil Majesty ES-2, Kuraray Noritake Dental Inc., Nagoya, Japan, (CM);-Harmonize Universal, Kerr Ltd., Uxbridge, UK, (HU);-PS2 experimental composite developed by Raluca Ripan Institute for Research in Chemistry, Babeș-Bolyai University, Cluj-Napoca, Romania, (PS2).

Two sample types were molded using Teflon dies for the specific test conducted in the current research. The first sample type was represented by standard discs of 20 mm diameter and 2 mm thickness (*n* = 60, 20/each material) for surface conditioning investigation, specifically roughness monitoring and color stability analysis. The second sample type was a standard cylindrical specimen of 4 mm diameter and 6 mm height for compression test (*n* = 60, 20/each material). Each of the samples was photo-polymerized for 20 s using a photo-polymerization lamp: Woodpecker LED B, Beijing, China. 

The same polishing and lustering protocol was applied for each sample. The plane surfaces of the photo-polymerized samples were ground with sandpaper with granulations of 800, 1000, and 2000 and further polished. Polishing procedures were carried out on SOF-LEX (3M ESPE, St. Paul, MN, USA, ABD) discs at 20,000 rpm followed by luster procedures using Crosshiny (Micerium, Genova, Italy) luster pastes in three successive stages: Shiny A with diamond particles of 3 μm; Shiny B with diamond particles of 1 μm; and Shiny C with colloidal alumina applied on a felt disc at increasing rotational speed and proper water humidification.

The samples’ initial height and mass were measured and determined before and after polishing. Thus, the height and mass losses were calculated with the following formulas, respectively:(1)$$H_{l}=\frac{H_{i}-H_{f}}{H_{i}}\times 100$$
(2)$$M_{l}=\frac{M_{i}-M_{f}}{M_{i}}\times 100$$
where *H_l_* and *M_l_* are the height and mass losses, respectively; *H_i_* and *M_i_* are the initial height and mass, respectively; and *H_f_* and *M_f_* are the final height and mass. respectively [25]. 

Sample morphology (both on initial and polished state as well as fractography imaging) was investigated by Scanning Electron Microscopy (SEM) using a microscope INSPECT S SEM (FEI Company, Hillsboro, OR, USA) operating in low vacuum mode at an acceleration voltage of 25 kV [26].

Atomic Force Microscopy (AFM) was carried out with a JEOL JSMP 4210 Scanning Probe Microscope produced by Jeol Company, Akashima, Japan. The samples surface was probed with NSC 15 cantilevers produced by MikroMasch Company, Sofia, Bulgaria, with a resonant frequency of 330 kHz and force constant of 40 N/m. The topographic images were scanned in tapping mode at a rate of 1 Hz. The images were further analyzed with JEOL WinSPM 2.0 processing software (Jeol Company, Akashima, Japan) by measuring the surface roughness *Ra* and *Rq* described by the following equations, respectively:(3)$$Ra=\frac{1}{l_{r}}\int_{0}^{l_{r}}\left|z(x)\right|dx$$
and
(4)$$Rq=\sqrt{\frac{1}{l_{r}}\int_{0}^{l_{r}}\left|z{\left(x\right)}^{2}\right|dx}$$
where *l* is the profile length and *z* is the height at *x* point. Both *Ra* and *Rq* are important for various research applications [21,22,25]. At least three different macroscopic areas were investigated on each sample and the mean roughness was calculated.

The mechanical properties were measured using a Lloyd LR5k Plus dual-column mechanical testing machine (Ametek/Lloyd Instruments, Meerbusch, Germany) with a maximum load of 5 kN. For each composite, three un-polished and polished specimens were tested. Compression strength results were statistically analyzed using the Anova Test followed by Tukey’s post hoc test on Origin Lab 2019 software produced by OriginLab Corporation, Northampton, MA, USA.

The samples’ color was investigated with a non-contact dental spectrophotometer VITA Easyshade Advance 4.0 (VITA Zahnfabrik H. Rauter GmbH & Co. KG, Bad Sackingen, Germany) using CIELab parameters. Eighteen specimens were included, which were split into 6 groups of two specimens that were stored in water for 24 h. The initial parameters were measured on a white background for three samples. Afterwards, two samples were immersed for 4 h in coffee infusion (10 g to 100 mL of water) and two other samples were immersed for 4 h in green tea infusion (5 g to 100 mL water). The samples were cleaned and stored in artificial saliva for 24 h at 37 °C. These operations were repeated daily for 5 days until the measurements. The measurements were realized using a colorimeter based on the CIELAB L*a*b* color space system, while the color differences (ΔE) were calculated based on the values before and after immersion, using the formula ΔE = [(ΔL*)^2^  +  (Δa*)^2^  +  (Δb*)^2^]^1/2^, where ΔE represents the total color difference and ΔL*, Δa*, and Δb* represent the changes in lightness, red–green coordinates, and yellow-blue coordinates, respectively. Final measurements were recorded after 5 days, for two different positions of the specimen, and the average was calculated for each specimen [27].

Sample size calculation

Given the paucity of previous studies on the topic, this randomized, controlled, in vitro study considered a sample size of *n* = 20 per group. Therefore, a statistical power of 0.8 was used, running a one-tailed test at a 5% significance level using G3*Power calculation software version 3.1.9.6. 

Statistical analysis

To check the distribution of continuous variables and to study their compliance with a normal distribution, basic descriptive statistics were used and the Shapiro–Wilk test of normal distribution was performed. Data were statistically analyzed using the Anova Test followed by Tukey’s post hoc test on Origin Lab 2019 software produced by OriginLab Corporation, Northampton, MA, USA. 

## 3. Results

One of the most important conditions for frontal restoration is the preparation of the material surface to ensure a coherent ensemble with the natural tissue of the tooth and to present similar characteristics. After molding the samples, a microstructure that is convolved between the mold surface topography and the composite constituents pressed onto these features is present. Therefore, the general aspect of the molded sample surface is slightly irregular, as observed by the SEM image in Figure 1. 

The mold surface asperities interact with the nanohybrid filler within CM, generating a wavy surface (Figure 1a). The HU sample contains only a nanostructural filler system that is uniformly embedded onto the superficial waves within the specimen surface, as shown in Figure 1b. The PS2 sample shows large microparticles partly uncoated by the polymer matrix surrounded by the nanofiller particles that are well distributed and embedded in the polymeric matrix, assuring a compact structure but with a rough aspect (Figure 1c). 

The CM sample after polishing and lustering procedures presents a very nice and smooth surface (Figure 1d), revealing the nanohybrid aspect of the composite. The SEM image in Figure 1d proves the good cohesion between polished microfiller particles and the polymeric matrix without excoriations or local delamination.

The HU sample after the polishing and lustering procedure is very smooth and uniform (Figure 1e), the wavy structure induced by the mold is completely removed, and the microstructural features are clearly visible. Figure 1e reveals only one small superficial defect on the upper-left side of the observation field where a 3 μm filler cluster is observed along with a partial delamination induced by the polishing procedures. 

Figure 1f reveals the barium oxide glass particles with polyhedral shapes and sizes ranging from 20 to 30 μm. These are surrounded by a compact nanofilled matrix structure. The outermost barium oxide particles are abraded by the polishing and lustering procedures, but they are still well embedded in the base matrix without delamination traces. The filler particle abundance gives a mosaic aspect of the microstructure which has a well-uniform, smooth surface.

The topographic images provided by the AFM investigation of the nanohybrid composites are presented in Figure 2. The CM topography (Figure 2a) reveals nanofiller particles of about 60 nm in diameter agglomerated into small submicron clusters with a rounded shape and sizes ranging from 300 to 600 nm that are uniformly distributed in the polymer matrix. This mixture is very well attached onto the barium glass particles situated on the central right side of the observation field in Figure 2a. Overall, this nanohybrid structure after polishing and lustering procedures assures a low value of the surface roughness of about 4 nm, as observed in Figure 3a. 

The HU composite presents a uniform topography with the nanofiller very well distributed in the polymer matrix (Figure 2b). The filler nanoparticles are about 40–60 nm and grouped in small, rounded submicron clusters of 150–300 nm. 

Only several clusters of about 400 nm occur randomly on the composite surface. The lack of micron particles allows a very advanced surface finishing with a roughness of about 2.5 nm (Figure 3a). Three roughness parameters *Ra, Rq*, and *Rz* are measured on the AFM images in Figure 2. Their variation plotted in Figure 3 shows that the *Ra* values are slightly lower due to being calculated differently using the arithmetic mean of the surface heights (Figure 3a). *Rq* is slightly increased because of the root-mean-square calculation of the heights (Figure 3b). Finally, *Rz* represents the maximum peak to valley height of the 3D profile and thus the obtained values are significantly greater (Figure 3c). All measured roughness parameters show that HU composite has a smoother surface while the PS2 surface is rougher due to the complex filler system.

The experimental nanohybrid composite PS2 presents a complex topography (Figure 2c) induced by the silica nanoparticles of about 40 nm in diameter and the hydroxyapatite nanoparticles of about 60 nm in diameter that are organized in a dense structure of submicron clusters of about 400–600 nm, well embedded in the polymer matrix closely surrounding the BaO_2_ microfiller particles. Such small barium oxide particles of about 3 μm are clearly visible in the center of the observation field of the image in Figure 2c. 

The results are displayed in Figure 4.

The HU nanohybrid composite presents a significant height loss compared to the composites with a complex filling system such as CM and PS2, as observed in Figure 4a. The CM and PS2 composites present a significantly higher mass loss than that observed for HU (Figure 4b). 

The results of the comparative compression test on both un-polished and polished samples that produced the curves are presented in Figure 5.

The values presented in Figure 6 reveal that un-polished specimens have a significantly lower compression strength than the polished samples for each of the tested composites, a fact sustained by statistical analysis where the calculated *p* values are below the significance threshold of 0.05, indicating significant differences. 

The statistical analysis on the data in Figure 6 reveals that the composites with the complex filler system present similar compressive strength values for CM and PS2 composites (*p* > 0.005). The HU composite presents lower compressive strength values compared to CM and PS2 (*p* < 0.05).

The general aspect of the composite failure under compression stress was observed with SEM at low magnification (100×) (Figure 7a–c). The CM and PS2 specimens present a spatial network of cracks developed on the larger filler particle alignments (Figure 7a,c), while the HU specimen shows failure that is promoted and developed within the polymer matrix without collateral crack development (Figure 7b). 

The sharp edges of the microfiller were tensioned, leading to local delamination of the larger particles and acting as failure initiators along with increasing solicitations (Figure 7d,f), while the rounded particles such as the nanofiller were less affected by the local deformations and were easily displaced along with the flowing polymer matrix (Figure 7e). 

The composites’ color was investigated using CIELab methods. The initial samples were firstly observed under a white background and were then subjected to repeated immersion in green tea and coffee (Figure 8) and analyzed. The best color stability was observed in the case of HU samples, while the experimental composite presented medium values (Figure 9). 

## 4. Discussion

Nanohybrid dental composites play a key role in the frontal restorations of teeth, fulfilling all therapeutic objectives by correlating their microstructure with physical, mechanical, optical, and biological requirements to ensure conditioning for the patient’s treatment. The present study aimed to investigate the characteristics of three dental composites designed for anterior restorations: a commercial nanohybrid spherical composite, a commercial nanohybrid composite, and an experimental nanohybrid composite. The null hypothesis was rejected, as the filler particles’ shape and size influence the mechanical properties and surface roughness, as well as color stability. 

The general aspect of the molded samples’ surface was analyzed using SEM and was slightly irregular. The waves resulting from the interaction between the mold and the composite material embed mainly the nanostructural filler and surround the microfiller particles which are partly exposed without the polymeric coating. The local irregularities are mainly generated by the mold characteristics and are less affected by the composite microstructure. The high cohesion between micro- and nanofiller systems within nanohybrid composite PS2 prevents the formation of the wavy features on the molded surface. These aspects clearly evidence that the molding process does not assure a proper surface for the frontal restorations. Therefore, all samples must be polished and lustered using the same preparation regime as described in the Materials and Methods section. In CM samples, large micro-particles of barium glass appear, with boulder aspects with some sharp corners that have a size range of 5–50 μm, which are very well distributed in the composite bulk, in good agreement with literature data [24,28]. These particles are very well embedded in the nanocomposite mass, assuring a coherent material. The polishing procedure abrades the top part of the superficial exposed microfiller particles, realizing an advanced surface smoothness. In the case of HU samples, the nanofiller particles are very well distributed in the polymer matrix, resulting in homogeneous and compact materials, and only some local filler clusters occur with rounded shapes and sizes of about 1 μm. The HU composite’s uniformity and high finishing onto the smooth surface are reported in the literature [29,30] because of the nanoparticles’ silanization, which facilitates the superficial wetting and subsequent composite compactness.

PS2 has a complex filler system based on BaO_2_ glass microparticles followed by the silica and hydroxyapatite nanoparticles as the nanofiller. All these granular materials were silanized prior to introduction into the polymeric matrix. Thus, the polymer mixture properly wets the particle surface, embedding them into a compact and coherent material.

The AFM investigation has two benefits: it allows proper visualization of the embedding of nanofiller particles and it can be used to measure the surface roughness. The nanohybrid structure of CM composite samples after polishing and lustering procedures assures a low value of the surface roughness of about 4 nm. This value is far below that of natural healthy human tooth enamel [22,31], proving the superior finishing properties of the composite. The HU composite presents a uniform topography with the nanofiller being very well distributed in the polymer matrix. The experimental nanohybrid composite PS2 presents a complex topography, induced by the silica nanoparticles and hydroxyapatite nanoparticles that are organized in a dense structure of submicron clusters, well embedded in the polymer matrix closely surrounding the BaO_2_ microfiller particles. This reported dense distribution of filler particles in the polymer matrix induced a slightly increased roughness, which optimally corresponds to the roughness of healthy natural tooth enamel [22,31]. Overall, PS2 presents the most compact nanostructure due to the optimal filler particle embedding, and the HU sample presents some local delamination occurring only for larger submicron clusters.

The results showed a lack of microfiller particles in the HU nanohybrid composite, which facilitates a significant height loss compared to the composites with complex filling systems such as CM and PS2. Larger particles of barium glass within CM and BaO_2_ glass in PS2 prevent a significant height reduction due to their internal resistance to abrasion. The CM and PS2 composites are denser than HU due to the significant amount of microfiller particles, a fact which influences the mass loss which is significantly larger than that observed for HU. Similar results were obtained by Garoushi et al. [25] in a study that analyzed the influence of nanometer-scale particulate fillers on some properties of microfilled composite resin. The results showed that the lowest surface roughness (Ra) was found in the group with 30% nanofillers and the roughest surface (Ra) was found in the group without nanofillers.

The significant microstructural differences between the composites’ surface prior to and after the polishing procedures might influence the restoration mechanical properties. Un-polished specimens have a significantly lower compression strength than polished samples for each of the tested composites. The compressive strength obtained for CM-polished samples is in good agreement with the data in the literature [32], but the values obtained for HU samples are slightly below the values mentioned in the literature [33]. The wavy structures observed on the composites’ unpolished surfaces that embed large micro-particles that are partially exposed on the outermost layer act as tension concentrators during pressing, acting as cracks initiators. This compressive effort is concentrated on these initial cracks, promoting in-depth failure, explaining the lower compressive strength obtained on the unpolished samples. The polished surfaces resist as a single monolith under compressive solicitations and uniformly dissipate the effort within the whole microstructure [33].

The composites with the complex filler system present similar compressive strength values. The HU composite presents lower compressive strength values compared to CM and PS2, as well as an average amount of filler and its distribution only at the nanostructural level, along with some clusters presenting local partial delamination.

The general aspect of composite failure under compression stress was observed with SEM at low magnification. The compressive stress was uniformly applied to the samples through the polished surface and was received by the micro- and nanostructural components. The CM and PS2 specimens presented a spatial network of cracks developed on the larger filler particle alignments, while the HU specimen evidenced a failure that was promoted and developed within the polymer matrix without collateral crack development. CM and PS2 deal with brittle failure when the specimen resists high loads and suddenly breaks down due to the almost-instant crack development, a fact that is in good agreement with the observation in the literature [34].

The main solicitation compresses the composites that generates an internal response, which implies that the lateral material flow is supported by the polymeric matrix observed by the samples’ height reduction and diameter swelling; thus, the filler particles require local repositioning regarding material flow. The sharp edges of the microfiller are tensioned, which leads to local delamination of the larger particles and acts as failure initiators along with increasing solicitations, while the rounded particles such as nanofillers are less affected by the local deformations and are easily displaced along with the flowing polymer matrix [35]. 

Besides the mechanical strength of the frontal restoration, its aesthetics is also very important. The CM samples present the lowest color stability, similar to previously reported literature data [36]. 

The factors that influence the color stability of the dental composites are represented by the resin matrix, the percentage of filler particles, the adsorption and absorption mechanisms of staining agents, as well as the interactions between composites and the staining agent. The composition of the resin materials and the relative amount of resin and filler content represent key factors in long-term color stability. Low-filler-content-resin-based materials have a reduced color stability due to increased water sorption [37]. Small particle fillers in composite resins determine a high color stability, compared to composites with large filler particles; this explains our result showing that HU is more stain-resistant than is CM.

Additionally, the water is absorbed directly in the resin matrix. A high amount of absorbed water can determine dimensional and morphological changes of the resin, improving the longevity of the resin but creating micro-cracks. Through these interfacial gaps or microcracks at the interface between the matrix and filler, the staining agent penetrates into the materials and discoloration appears [38,39].

The polishing procedure also influences the coloring ability of composite dental materials. In the case of nanohybrid composites, studies indicate that during the finishing and polishing procedures, smaller filler particles are removed, and small voids remain at the surface of the restorative material, allowing the staining agents to penetrate more easily [40]. A study by Nasim et al., which evaluated the color stability of microhybrid, nanohybrid, and microfilled composite resins, reported that microhybrid composite resins have a more stable color compared to nanohybrid and microfilled composites. This might be due to the resin matrix nature and potential porosity in the aggregated filler particles as well as to the porosity of the barium glass fillers [41].

The real innovation that implies better mechanical behavior is the nanofiller’s possibility of improving the load of the inorganic phase when compared to microfilled composites. 

Clinically, the type of composite resin used for dental reconstruction as well as the polishing procedure influences the aesthetics and long-term resistance. In the case of aesthetic appearance, some studies investigated the correlation between the dental composite gloss and roughness of the restorations and concluded that a perfectly polished surface corresponds to a smooth surface, which exhibits clinical durability and satisfactory aesthetics [42,43]. At the same time, a rough and irregular restoration surface can be easily affected by superficial stains and plaque accumulation [44], leading to gingival inflammation and secondary caries [45].

The current study presents several limitations, including all factors related to the type of research, which is in vitro. Some important drawbacks are present, such as the comparison to an in vivo study in the oral cavity, a complex environment in which saliva dilutes the solutions used and changes their pH; in addition, it contains various enzymes and effective salts. Restoration materials (e.g., composite resins) are exposed to a wide range of thermal changes following the consumption of hot and cold foods and beverages. The physical properties of materials also change over time. Additionally, other variables can have a significant influence on composite behavior, such as cure depth [46] and fiber incorporation [47]. Also, these variables should be included in future studies.

## 5. Conclusions

The investigated nanohybrid composite behavior strongly depends on the microstructural features. The presence of both nano- and microparticles as a complex filler system within CM and PS2 improves the wear and compressive strengths compared to the simple nanofiller in HU. CM and PS2 have brittle failure that occurs at high load, while HU presents a more tenacious failure with significant local deformations supported by the polymer matrix. Polishing and lustering procedures are very important to preserve the proper mechanical behavior of the composite dental materials for anterior restorations.

## Figures and Tables

**Figure 1 biomedicines-12-00243-f001:**
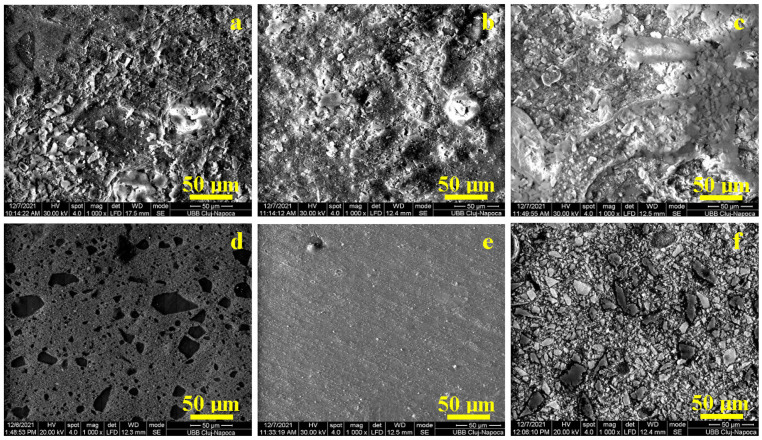
SEM images for the samples’ surface after molding: (**a**) CM, (**b**) HU, and (**c**) PS2; and after lustering: (**d**) CM, (**e**) HU, and (**f**) PS2. All images are taken at the same magnification 1000×.

**Figure 2 biomedicines-12-00243-f002:**
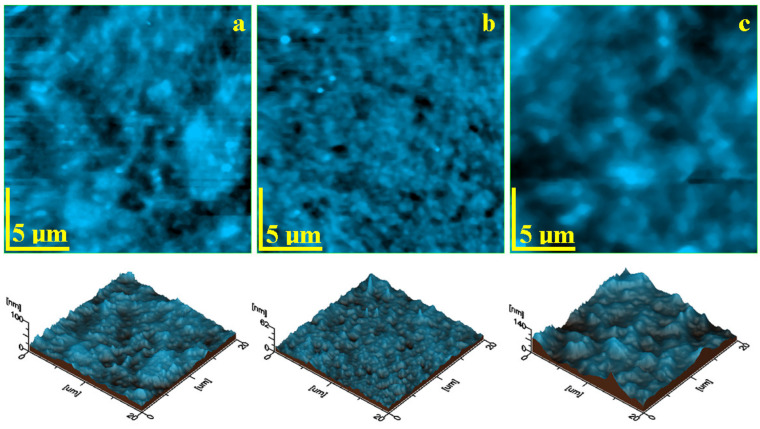
AFM topographic images for the sample surface after lustering: (**a**) CM, (**b**) HU, and (**c**) PS2. The 3D profiles are given below each topographic image.

**Figure 3 biomedicines-12-00243-f003:**
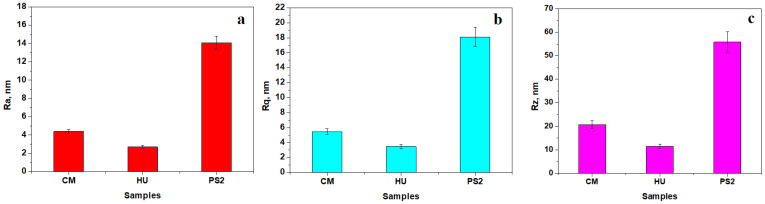
Surface roughness for the composite samples after polishing and lustering procedures: (**a**) *Ra*, (**b**) *Rq*, and (**c**) *Rz*. Error bars represent standard deviation.

**Figure 4 biomedicines-12-00243-f004:**
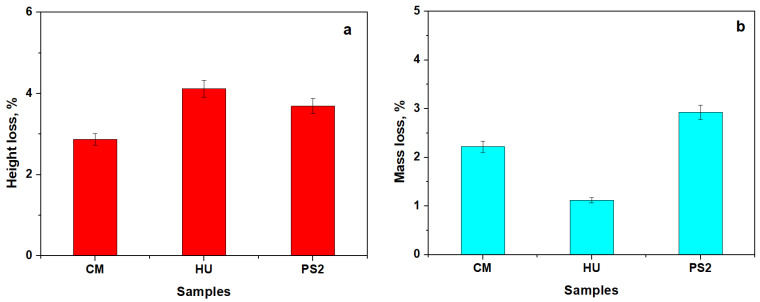
Samples’ height loss (**a**) and samples’ mass loss (**b**) before and after polishing. Error bars represent the standard deviation.

**Figure 5 biomedicines-12-00243-f005:**
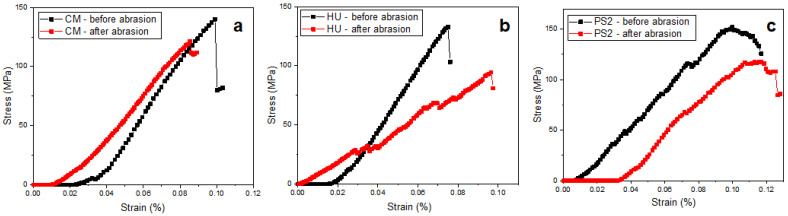
Experimental curves obtained at compression test for (**a**) CM, (**b**) HU, and (**c**) PS2.

**Figure 6 biomedicines-12-00243-f006:**
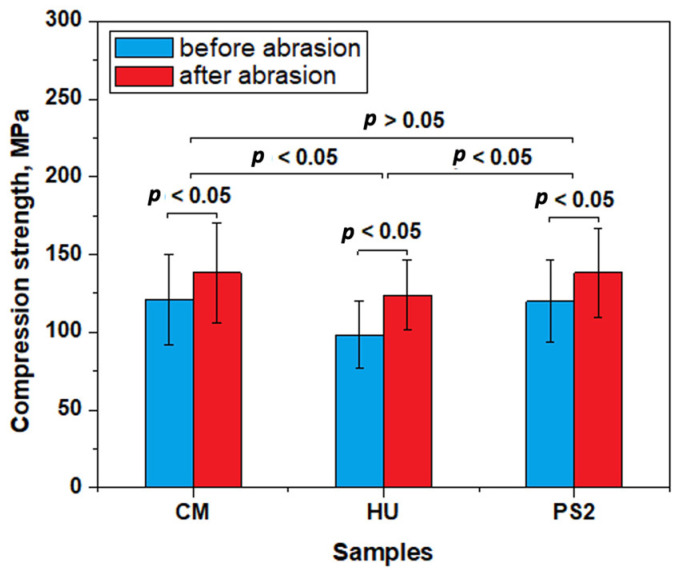
Compression strength before and after abrasion test. Error bars represent standard deviation.

**Figure 7 biomedicines-12-00243-f007:**
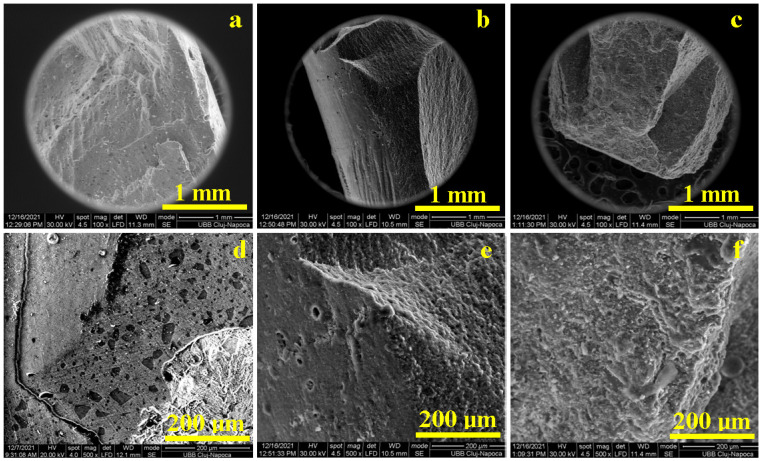
SEM fractography for the composite samples at low magnification (100×): (**a**) CM, (**b**) HU, and (**c**) PS2; and high magnification (500×): (**d**) CM, (**e**) HU, and (**f**) PS2.

**Figure 8 biomedicines-12-00243-f008:**
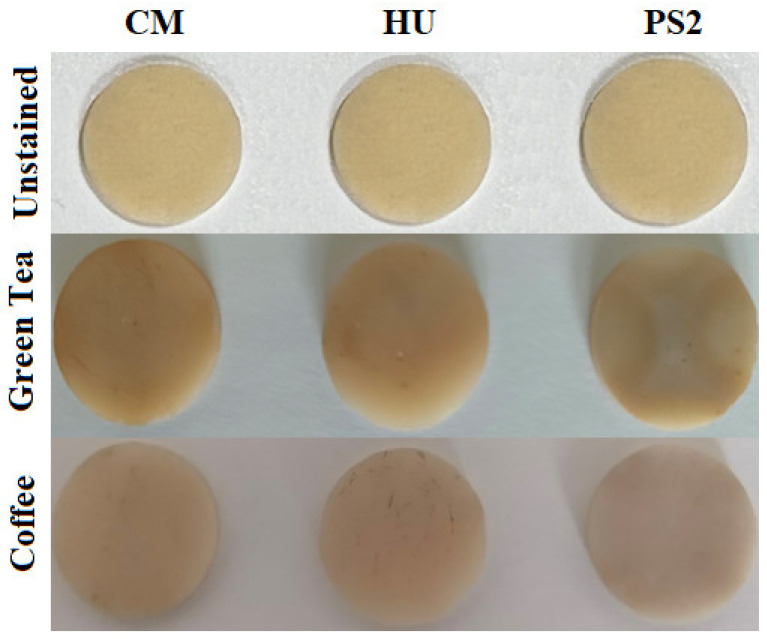
Composite samples subjected to CIELab color test.

**Figure 9 biomedicines-12-00243-f009:**
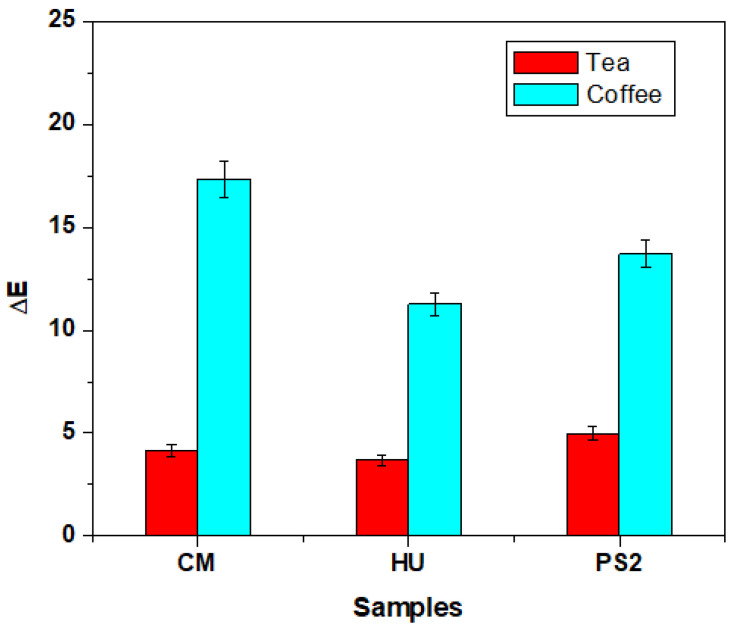
Composite color difference induced by immersion in green tea and coffee evidenced by CIELab test. Error bars represent standard deviation.

**Table 1 biomedicines-12-00243-t001:** Materials included in this study.

Code	Material	Type	Matrix	Filler
HU	Harmonize Universal (Kerr Ltd., Uxbridge, UK)	Nanohybrid spherical	Bis-GMA, TEGDMA, Bis-EMA	5–400 nm particle size. Spherical zirconia and silica nanoparticles, rheological modifier.
CM	Clear Fil Majesty ES-2 (Kuraray Noritake Dental Inc., Nagoya, Japan)	Nanohybrid	Bis-GMA, hydrophobic aromatic DMA, and hydrophobic aliphatic DMA, dl-camphorquinone	Silanated barium glass (particle size 0.37–1.5 μm) and pre-polymerized organic filler.
PS2	Experimental composite resin	Nanohybrid	Bis-GMA, UDMA, TEGDMA	40–60 nm particle size. HA-Zn, colloidal silica, barium glasses, lantan glasses

## Data Availability

Data are available upon request from the corresponding authors.

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
