# Peer review of "Properties of Nanohybrid Dental Composites—A Comparative In Vitro Study"

_biomedicines, 2024, doi:10.3390/biomedicines12010243_

Round 1

Reviewer 1 Report

Comments and Suggestions for Authors

Dear authors, congratulations for the interesting work you've done. The method that you used to compare different composites is impeccable and rigorous and the descriptions of results are complete. Neverthless, the topic is not novel because it is well ascertained that the composite filler particles shape and size does influence their mechanical and aesthetical properties. However, you introduce a new experimental composite, so it could be taken into consideration. 

Here are a few suggestions to improve your paper.

- 28 references only in introduction are too many. Please reduce the references in the introduction

- Please specify the particles diameter of PS2 to allow a rapid comparation to the reader

- Discussion is complete and well organized, but you have to analyze better the experimental composite and its further applications.

Hope you find it useful

Author Response

Dear reviewer,

Thank you for your very helpful advice and guidelines regarding manuscript preparation. We have read your indications and the following modifications have been made to the manuscript:

  1. We have reduced the number of references in the introduction.
  2. The size of the PS2 particles has been added.
  3. The discussion section has been modified according to the indications.

Reviewer 2 Report

Comments and Suggestions for Authors

The paper entitled „Nanohybrid dental composites properties - a comparative in-vitro study” focuses on the investigation of three dental composites designed for frontal restorations: a commercial nanohybrid spherical composite; a commercial nanohybrid composite and an experimental nanohybrid composite and to reveal their micro and nano structural aspects.  The topic of the paper is interesting. Although the introduction can be improved, the results are well illustrated and explained. However, I would like to recommend a major revision of the paper for publication because of the following main issues:

1.               The introduction section should be elaborated. Instead of many methodological details, the authors can give more quantitative results;

2.               In the introduction, more studies focused on similar topics could be better revealed so the reader to understand the recent achievements in the field.

3.               The equations in the text should be subsequently numbered;

4.               Both Ra and Rq have very similar values, It would be more informative for the reader the authors to show for example Rz or Rsk values from the AFM measurements.

5.               The abrasion tests mentioned do not follow a particular standard. The methodology allows for the influence of random factors such as pressure force, polishing time, etc., which can significantly change the results. For that reason, in my opinion, the presented abrasion test results are not representative

Comments on the Quality of English Language

None

Author Response

Thank you for your very helpful advice and guidelines regarding manuscript preparation. We have read your indications and the following modifications have been made to the manuscript:

  1. In the introduction, more studies focused on similar topics could be better revealed so the reader to understand the recent achievements in the field. 

R: Supplementary information in the introduction has been added.

  1. The equations in the text should be subsequently numbered;

R: The equations in the text were subsequently numbered

  1. Both Ra and Rq have very similar values, It would be more informative for the reader the authors to show for example Rz or Rsk values from the AFM measurements. 

R: We added Rz variation beside Ra and Rq, please see the revised Figure 3. Unfortunately the analysis software do not allows us measuring Rsk on the obtained AFM images. The text was completed with additional discussion of Figure 3.

  1. The abrasion tests mentioned do not follow a particular standard. The methodology allows for the influence of random factors such as pressure force, polishing time, etc., which can significantly change the results. For that reason, in my opinion, the presented abrasion test results are not representative. 

R: The authors consider relevant the results obtained as they reflect the clinical behavior of these materials after polishing. Indeed, it does not follow the standard protocol for the abrasion test. Consequently, all statement was rephrased. 

Reviewer 3 Report

Comments and Suggestions for Authors

Dear Authors,

I have read the manuscript with interest and some questions raised. Enlisted please find my comments.

Overall. General English grammar revision (Minor spelling errors).

Key words. “dentistry” and “restorative dentistry” could be added in my opinion.

Abstract. Please add the names of the statistical tests in this section.

Introduction. Authors stated “Filler effect is to enhance the microstructural aspects of the composite that further improves the mechanical properties. Some of the composite resins are designed only with micro or nano-structured fillers but the newest trends for the dental composites use both micro and nano fillers generating complex systems that define the nanohybrid structures”. Please add a reference for this statement.

Materials and Methods. Please add if and how sample size calculation has been performed.

Materials and Methods. Please add a reference for each method.

Materials and Methods. For each material used, please add details about commercial name manufacturer, City and State.

Materials and Methods. For each machinery used, please add details about commercial name manufacturer, City and State.

Materials and Methods. Please add details about software used, version, Manufacturer, City and State.

Materials and Methods. Authors stated “Compression strength results were statistically analyzed using Anova Test”. ANOVA is used for gaussian distributions. Please explain how normality of data was tested.

Materials and Methods. Authors stated “Tuckey post hoc test”. Please insert correct name “Tukey”.

Materials and Methods. Statistics. Please add significance level for P values (0.05? 0.01?).

Results. Please add P values all along this section.

Discussion. Authors stated “The polishing procedure also influences the coloring ability of composite dental materials. It has been reported that smaller filler particles are removed during polishing and finishing operations in nanohybrids, and small voids remain at the surface of the restorative material, making the material more susceptible to staining [41].”. Provide a general interpretation of the results in the context of other evidence, and implications for future research. It could be added that “Additionally,other variables can have a significant influence on composite behavior, such as Depth of cure (Colombo M, Gallo S, Poggio C, Ricaldone V, Arciola CR, et al. New Resin-Based Bulk-Fill Composites: in vitro Evaluation of Micro-Hardness and Depth of Cure as Infection Risk Indexes. Materials (Basel). 2020 Mar 13;13(6):1308. doi: 10.3390/ma13061308), fiber incorporation (Peker O, Bolgul B. Evaluation of surface roughness and color changes of restorative materials used with different polishing procedures in pediatric dentistry. J Clin Pediatr Dent. 2023 Jul;47(4):72-79. doi: 10.22514/jocpd.2023.037. Epub 2023 Jul 3.). Also these variables should be included in future studies”. These concerns should be added to Discussion section.

Tables. No tables with descriptive and inferential statistics have been presented. Please add some tables showing the results and descriptive statistics of the main variables tested.

Figure 1. Please add magnification.

Figure 3. Please add in the caption the meaning of error bars (Standard deviation? Standard error?)

Figure 4. Please add in the caption the meaning of error bars (Standard deviation? Standard error?)

Figure 5. Please enlarge the figures in order to increase readability.

Figure 6. Please add in the caption the meaning of error bars (Standard deviation? Standard error?)

Figure 7. Please add magnification.

Figure 9. Please add in the caption the meaning of error bars (Standard deviation? Standard error?)

Comments on the Quality of English Language

General English grammar revision (Minor spelling errors).

Author Response

Dear reviewer,

Thank you for your very helpful advice and guidelines regarding manuscript preparation. We have read your indications and the following modifications have been made to the manuscript:

  1. Key words. “dentistry” and “restorative dentistry” could be added in my opinion.

The keywords „dentistry” and „restorative dentistry” have been added.

  1. Please add the names of the statistical tests in this section.

R: Information added.

  1. Authors stated “Filler effect is to enhance the microstructural aspects of the composite that further improves the mechanical properties. Some of the composite resins are designed only with micro or nano-structured fillers but the newest trends for the dental composites use both micro and nano fillers generating complex systems that define the nanohybrid structures”. Please add a reference for this statement.

R: Reference was added.

  1. Materials and Methods. Please add if and how sample size calculation has been performed. 

R: Information on sample size calculation was added.

  1. Materials and Methods. Please add a reference for each method.

R: Reference added.

  1. Materials and Methods. For each material used, please add details about commercial name manufacturer, City and State. 

R: Information added.

  1. Materials and Methods. For each machinery used, please add details about commercial name manufacturer, City and State. 

R: Information added.

  1. Materials and Methods. Please add details about software used, version, Manufacturer, City and State.

R: Detalis added

  1. Materials and Methods. Authors stated “Compression strength results were statistically analyzed using Anova Test”. ANOVA is used for gaussian distributions. Please explain how normality of data was tested.

R: Explanation was included in the text.

  1. Materials and Methods. Authors stated “Tuckey post hoc test”. Please insert correct name “Tukey”.

R: Corrected.

  1. Materials and Methods. Statistics. Please add significance level for P values (0.05? 0.01?).

R: Information added.

  1. Please add P values all along this section.

R: Values added.

  1. Authors stated “The polishing procedure also influences the coloring ability of composite dental materials. It has been reported that smaller filler particles are removed during polishing and finishing operations in nanohybrids, and small voids remain at the surface of the restorative material, making the material more susceptible to staining [41].”. Provide a general interpretation of the results in the context of other evidence, and implications for future research. It could be added that “Additionally,other variables can have a significant influence on composite behavior, such as Depth of cure (Colombo M, Gallo S, Poggio C, Ricaldone V, Arciola CR, et al. New Resin-Based Bulk-Fill Composites: in vitro Evaluation of Micro-Hardness and Depth of Cure as Infection Risk Indexes. Materials (Basel). 2020 Mar 13;13(6):1308. doi: 10.3390/ma13061308), fiber incorporation (Peker O, Bolgul B. Evaluation of surface roughness and color changes of restorative materials used with different polishing procedures in pediatric dentistry. J Clin Pediatr Dent. 2023 Jul;47(4):72-79. doi: 10.22514/jocpd.2023.037. Epub 2023 Jul 3.). Also these variables should be included in future studies”. These concerns should be added to Discussion section.

R: Information added.

  1. Figure 1. Please add magnification.

R: The magnification was added.

  1. Figure 3. Please add in the caption the meaning of error bars (Standard deviation? Standard error?)

R: Thank you, the error bars are standard deviation. The caption was revised accordingly.

  1. Figure 4. Please add in the caption the meaning of error bars (Standard deviation? Standard error?)

R: Thank you, the error bars are standard deviation. The caption was revised accordingly.

  1. Figure 5. Please enlarge the figures in order to increase readability.

R: Figure 5 was enlarged accordingly.

  1. Figure 6. Please add in the caption the meaning of error bars (Standard deviation? Standard error?)

R: Thank you, the error bars are standard deviation. The caption was revised accordingly.

  1. Figure 7. Please add magnification.

R: The magnification was added.

  1. Figure 9. Please add in the caption the meaning of error bars (Standard deviation? Standard error?)

R: Thank you, the error bars are standard deviation. The caption was revised accordingly.

Round 2

Reviewer 3 Report

Comments and Suggestions for Authors

All comments have been answered thank you 

Comments on the Quality of English Language

Authors revised the text and in my opinion it is suitable for publication